# On Robustness of Finetuned Transformer-based NLP Models

**Pavan Kalyan Reddy Neerudu**[1]*, **Subba Reddy Oota**[2]*, **Mounika Marreddy**[3]
**Venkateswara Rao Kagita**[1], **Manish Gupta**[3,4]
[1]NIT Warangal, India; [2]INRIA, Bordeaux, France; [3]IIIT Hyderabad, India; [4]Microsoft, India
neerud_951963@student.nitw.ac.in, subba-reddy.oota@inria.fr,mounika.marreddy@research.iiit.ac.in
venkat.kagita@nitw.ac.in, gmanish@microsoft.com

## Abstract

Transformer-based pretrained models like BERT, GPT-2 and T5 have been finetuned for a large number of natural language processing (NLP) tasks, and have been shown to be very effective. However, while finetuning, what changes across layers in these models with respect to pretrained checkpoints is under-studied. Further, how robust are these models to perturbations in input text? Does the robustness vary depending on the NLP task for which the models have been finetuned? While there exists some work on studying the robustness of BERT finetuned for a few NLP tasks, there is no rigorous study that compares this robustness across encoder only, decoder only and encoder-decoder models.

In this paper, we characterize changes between pretrained and finetuned language model representations across layers using two metrics: CKA and STIR. Further, we study the robustness of three language models (BERT, GPT-2 and T5) with eight different text perturbations on classification tasks from the General Language Understanding Evaluation (GLUE) benchmark, and generation tasks like summarization, free-form generation and question generation. GPT-2 representations are more robust than BERT and T5 across multiple types of input perturbation. Although models exhibit good robustness broadly, dropping nouns, verbs or changing characters are the most impactful. Overall, this study provides valuable insights into perturbation-specific weaknesses of popular Transformer-based models, which should be kept in mind when passing inputs. We make the code and models publicly available[1].

## 1 Introduction

Pretrained Transformer-based language models (Vaswani et al., 2017) have revolutionized the field of natural language processing (NLP). Specifically, pretrained Transformer models such as BERT (Devlin et al., 2018) (Bidirectional Encoder Representations from Transformers), GPT-2 (Radford et al., 2019) (Generative Pre-trained Transformer) and T5 (Raffel et al., 2020) (Text-To-Text Transfer Transformer), have set new benchmarks for various downstream NLP tasks. To adapt these models for downstream tasks, researchers finetune them on task-specific data. Finetuning modifies the representations generated by each layer. How do these modifications compare with respect to corresponding pretrained models?

Further, if we perturb the inputs supplied to these language models, does that lead to changes in layer representations and also prediction accuracy? How does this robustness vary across different NLP tasks for which these models have been finetuned? It is important to understand answers to these questions to ensure that we account for perturbations for which these models are not very robust.

Recent investigations on adversarial text perturbations (Wang et al., 2021; Jin et al., 2020; Li et al., 2020; Garg and Ramakrishnan, 2020; Sanyal et al., 2022) have revealed that even strong language models are susceptible to adversarial examples, which can increase the risk of misclassification of input data, leading to incorrect results. However, existing studies on robustness have used only a few adversarial text perturbations and experimented with BERT on a few downstream NLP tasks.

Hence, it is important to study the representations generated by pre-trained and finetuned Transformer models and evaluate their robustness when exposed to text perturbations. Specifically, we aim to answer the following questions: (i) Is the effect of finetuning consistent across all models for various NLP tasks? (ii) To what extent are these models effective in handling input text perturbations? and (iii) Do these models exhibit varying levels of robustness to input text perturbations when finetuned

---

* The first two authors made equal contribution.
[1]https://github.com/PavanNeerudu/Robustness-of-Transformers-models

for different NLP tasks?

Earlier studies use representation similarity analysis (RSA) as the metric to quantify the similarity between the representations generated by different models (He et al., 2021). Recently, Kornblith et al. (2019) introduced a new similarity metric, Centered Kernel Alignment (CKA), to better measure the representations. In another recent study, Nanda et al. (2022) overcame the limitations of CKA by introducing 'Similarity Through Inverted Representations' (STIR) to analyze the shared invariances of models to meaningless perturbations. Hence, we use both CKA and STIR to examine the representation similarity across various layers of pretrained versus finetuned models. We measure robustness as a function of relative change in performance when perturbations are applied to inputs. We make the code and models publicly available[1].

Our key contributions are as follows: (1) Our analysis of finetuned models versus pretrained models shows that the last layers of the models are more affected than the initial layers when finetuning. (2) GPT-2 exhibits more robust representations than BERT and T5 across multiple types of input perturbation. (3) Although Transformers models exhibit good robustness, the models are seen to be most affected by dropping nouns, verbs or changing characters. (4) We also observed that while there is some variation in the affected layers between models and tasks due to input perturbations, certain layers are consistently impacted across different models, indicating the importance of specific linguistic features and contextual information.

## 2   Related Work

The NLP community has shown increasing concern for the robustness of pre-trained language models, when exposed to adversarial examples. To assess this, various studies have been conducted to examine the models' susceptibility to modifications in the input text. Some works investigated modifications including typos or word replacements, while Li et al. (2020); Jin et al. (2020); Sun et al. (2020) evaluated the models' capacity to adapt to different data distributions and linguistic phenomena, such as coreference or sarcasm. To address the concern for robustness, Wang et al. (2021) introduced a multi-task benchmark for the evaluation of language models. More broadly, Schiappa et al. (2022) perform robustness analysis of video-language models.

Studies on model probing, such as (Tenney et al., 2019b; Liu et al., 2019; Tenney et al., 2019a; Hewitt and Manning, 2019), have analyzed the degree to which syntactic and semantic features are captured in the different layers of BERT-like models. Additionally, Zhou and Srikumar (2021); Merchant et al. (2020) performed a comprehensive analysis of how finetuning affects the representations in the BERT model using a combination of probing and analytical techniques.

In terms of similarity metrics, Voita et al. (2019) used a form of canonical correlation analysis (PW-CCA; (Morcos et al., 2018)) to examine the layer-wise evolution of representations in deep neural networks. On the other hand, Abnar et al. (2019) utilized Representation Similarity Analysis (RSA; (Laakso and Cottrell, 2000; Kriegeskorte et al., 2008)) to assess the models' representations. In recent work, Wu et al. (2020) applied Centered Kernel Alignment (CKA; (Kornblith et al., 2019)) to pre-trained Transformers like BERT and GPT-2, focusing mainly on cross-model comparisons.

Our study, however, specifically delves into comparing the representations generated by pre-trained and finetuned Transformer models, and analyzing their layer-wise similarity and shared invariances. Additionally, our work seeks to contribute to the ongoing efforts to better understand the strengths and limitations of Transformer models in handling input text perturbations.

## 3   Methodology

### 3.1   Representational Similarity Analysis between pretrained and finetuned models

We use the following two metrics for comparing pretrained and finetuned models: CKA and STIR. **CKA** We use CKA (Kornblith et al., 2019) to compare the layer-wise hidden state representations of pre-trained and finetuned models for each dataset. CKA=1 indicates perfect similarity while CKA=0 indicates no similarity. (Linear) CKA between input matrices $X$ and $Y$ is computed as $\text{CKA}(X,Y) = \frac{\text{HSIC}(K,L)}{\sqrt{\text{HSIC}(K,K)\text{HSIC}(L,L)}}$ where $K$ and $L$ are the similarity matrices computed as $K = XX^T$ and $L = YY^T$, and $\text{HSIC}(K,L)$, $\text{HSIC}(K,K)$, and $\text{HSIC}(L,L)$ are the Hilbert-Schmidt Independence Criterion (HSIC) values.

Specifically, we extract the hidden states corresponding to each token within a sentence for every language model. These hidden states are then averaged to create a unified 768D representation for

that sentence. For a 100 sentence input, we obtain [13, 100, 768] dimensional output (comprising 1 embedding and 12 layers) for each pre-trained as well as fine-tuned model. Then, we compute the CKA values at every layer (between the [100, 768] representations) yielding 13 layer-wise values for each combination of dataset and model.

**STIR** To evaluate the shared invariance between pre-trained and finetuned models, we use STIR. First, we obtain hidden state representations for the pre-trained models on the test dataset for all GLUE tasks. Then, we sample half of the dataset (except for QQP, where 5000 examples were used) 20 times and obtain $X'$ for $X$, where $X'$ represents the examples with the smallest L2 norm with respect to these representations. Finally, we compute the CKA similarity between the hidden state representations generated by the finetuned model and pre-trained model, and report it as **STIR(finetuned|pre-trained)**.

To measure the invariance in the opposite direction, we follow the same procedure with the finetuned model. We obtain hidden state representations for the finetuned model, find the examples with the smallest L2 norm, and use these examples' hidden state representations of the pre-trained model to find **STIR(pre-trained|finetuned)**.

As proposed in (Nanda et al., 2022), STIR values are computed as $\text{STIR}(m_2|m_1, X) = \frac{1}{k}\sum_{X'} \text{CKA}(m_2(X), m_2(X'))$ where $m_1$ and $m_2$ are the models under comparison, $X$ is the test dataset and $X'$ are similar examples obtained using the representation inversion method mentioned above. We fix $k$=20.

### 3.2 Text perturbations

We examine various types of text perturbations that encompass a wide range of variations that can occur in natural language text. They are defined as follows. (1) **Drop noun/verb** perturbations involve dropping words based on their part-of-speech tag, specifically nouns or verbs. (2) **Drop first/last** perturbations alter the phrase based on its location. Specifically, the first/last word is dropped. (3) **Swap text** perturbations involve swapping one or more words from the original phrase. (4) **Change char** perturbations involve changing one or more characters in a word(s). (5) **Add text** perturbations involve appending irrelevant word(s) to the text. (6) **Bias** perturbations involve switching the gender of one or more words in a phrase.

### 3.3 Representations and Tasks for Robustness Evaluation

We experiment with a Transformer encoder (BERT-base), a decoder (GPT-2) and an encoder-decoder model (T5-base) for classification tasks. For generative tasks, we use GPT-2 and T5-base. For obtaining model representations, we extract the hidden states for BERT and GPT-2 and encoder hidden states for T5 for each token and average them to obtain a single representation of 768 size.

For evaluation across classification tasks, we use General Language Understanding Evaluation (GLUE; (Wang et al., 2018)) benchmark, a collection of diverse NLP tasks. The tasks include: (1) **Single-sentence tasks**: The Corpus of Linguistic Acceptability (CoLA; (Warstadt et al., 2019)) and Stanford Sentiment Treebank (SST-2; (Socher et al., 2013)). (2) **Similarity and paraphrase tasks**: Microsoft Research Paraphrase Corpus (MRPC; (Dolan and Brockett, 2005)), Semantic Textual Similarity Benchmark (STS-B; (Cer et al., 2017)) and Quora Question Pairs (QQP; (Iyer et al., 2017)). (3) **Inference tasks**: Multi-Genre Natural Language Inference (MNLI; (Williams et al., 2017)) (with two splits: matched and mismatched), Question Natural Language Inference (QNLI; (Rajpurkar et al., 2016)), Recognizing Textual Entailment (RTE; (Dagan et al., 2005)), Winograd Natural Language Inference (WNLI; (Levesque et al., 2012)), and Abreviated eXperiments (AX).

For generative tasks, we evaluate text summarization, free-form text generation and question generation. For text summarization, we use the Extreme Summarization (**XSum** (Narayan et al., 2018)) dataset. XSum contains news articles accompanied by single-sentence summaries, providing a diverse range of topics and presenting challenges in extractive summarization. For free-form text generation, we utilized the **CommonGen** (Lin et al., 2019) dataset, which consists of sentence pairs serving as prompts for everyday scenarios. This evaluates the models' capability to generate coherent and contextually relevant descriptions. Regarding question generation, we leveraged the Stanford Question Answering Dataset (**SQuAD** (Rajpurkar et al., 2016)). It includes passages from various sources along with question-answer pairs. In this task, our models were tasked with generating questions based on input context and answer.

We use pre-trained BERT-base, GPT-2, and T5-base models from HuggingFace v4.2.2 (Wolf et al.,

2020). We finetune GPT-2 and T5 models on GLUE tasks mentioned above and obtain the finetuned models for BERT from HuggingFace. Further details on detailed task descriptions and metrics are in Appendix.

### 3.4 Evaluation Metrics for Robustness

For classification tasks, we used metrics like Matthews Correlation Coefficient for CoLA, Pearson Correlation Coefficient for STS-B and Accuracy for other tasks. For generative tasks, we employed ROUGE (Lin, 2004). We report ROUGE-1, ROUGE-2, and ROUGE-L F-scores.

Let $m_c$ and $m_p$ denote the values of a metric $m$ of the model on the clean and perturbed test sets, respectively. Then, we define robustness as robustness $= 1 - \frac{m_c - m_p}{m_c}$. Typically, robustness score of a model ranges between 0 (not robust) and 1 (very robust). Score greater than 1 suggests that the model's performance improves with the perturbation applied.

## 4 Results and Analysis

### 4.1 How does finetuning modify the layers representations for different models?

Fig. 1 shows the layer-wise CKA/STIR comparisons between pre-trained and finetuned BERT, GPT-2 and T5 models for the GLUE benchmark. We observe that the impact of finetuning varies across models. GPT-2's last layers were more affected than BERT's during the finetuning process, but GPT-2 had fewer affected layers, indicating higher semantic stability. CKA values for GPT-2 remained mostly higher than those for BERT, suggesting more general applicability of GPT-2's pre-trained representation. T5 showed similar effects of finetuning as BERT but with more pronounced effects, as evidenced by the CKA values.

All the models on the majority of datasets showed a gradual decrease in similarity of representations in the initial layers, with a larger drop in similarity observed later on. BERT and GPT-2 had the highest CKA values for the RTE and WNLI datasets, indicating that these models were least affected by finetuning. In contrast to BERT and GPT-2, T5's CKA values increased from layer 11 to 12 on some datasets. This could be due to the encoder-decoder architecture of T5, where the decoder is introduced after the 12th layer, causing the model to converge towards a more generalizable input representation.

Another noteworthy observation was the similar trend in CKA and STIR values for some datasets, which was consistent across all three models. This suggests that the underlying data characteristics of these particular tasks are better captured by the pre-trained representations of these models. This finding explains why transfer learning using these models is so successful, where these pre-trained models have been finetuned on other related tasks, resulting in improved performance due to the shared invariance of these representations. It was also observed that in some cases, CKA values were slightly higher compared to STIR values. This difference could be attributed to CKA overestimating the similarity between models.

**Layer-wise analysis**: For this experiment, we constructed a logistic regression model at each layer for both pre-trained and finetuned language models (e.g., BERT). These models were trained using the hidden state representations of `[CLS]` token generated by the corresponding BERT models. We then used the trained logistic regression models to infer the test dataset using the hidden state representations of `[CLS]` token generated by both pre-trained and finetuned BERT models. We repeat the experiment for GPT-2 also using the last token hidden state representations. Figs. 6 and 7 (in Appendix) show the relationship between CKA and layer-wise accuracy for BERT, and GPT-2 models. Fig. 8 (in Appendix) shows CKA/STIR plots for T5. For almost every task across all three models, we observe that the CKA values drop from initial to later layers. Our experiments reveal a correlation between CKA and accuracy, with a decrease in CKA values indicating a greater difference in accuracy between the pre-trained and finetuned models.

Overall, based on the above results, the following insights can be drawn. (1) **Task and model sensitivity:** Obviously performance across NLP tasks varies for each model. Each task has its own unique characteristics that finetuning captures. This clearly shows up in the results since the CKA/STIR values vary significantly across NLP tasks even for the same model. (2) **Layer influence:** The results also demonstrate that the impact of finetuning is not evenly distributed across all layers of various models. Later layers are seen to be more impacted than the lower layers. (3) **Layer-wise finetuning analysis:** The layer-wise analysis of models' performances sheds light on the inner workings of deep learning models and highlights the importance of

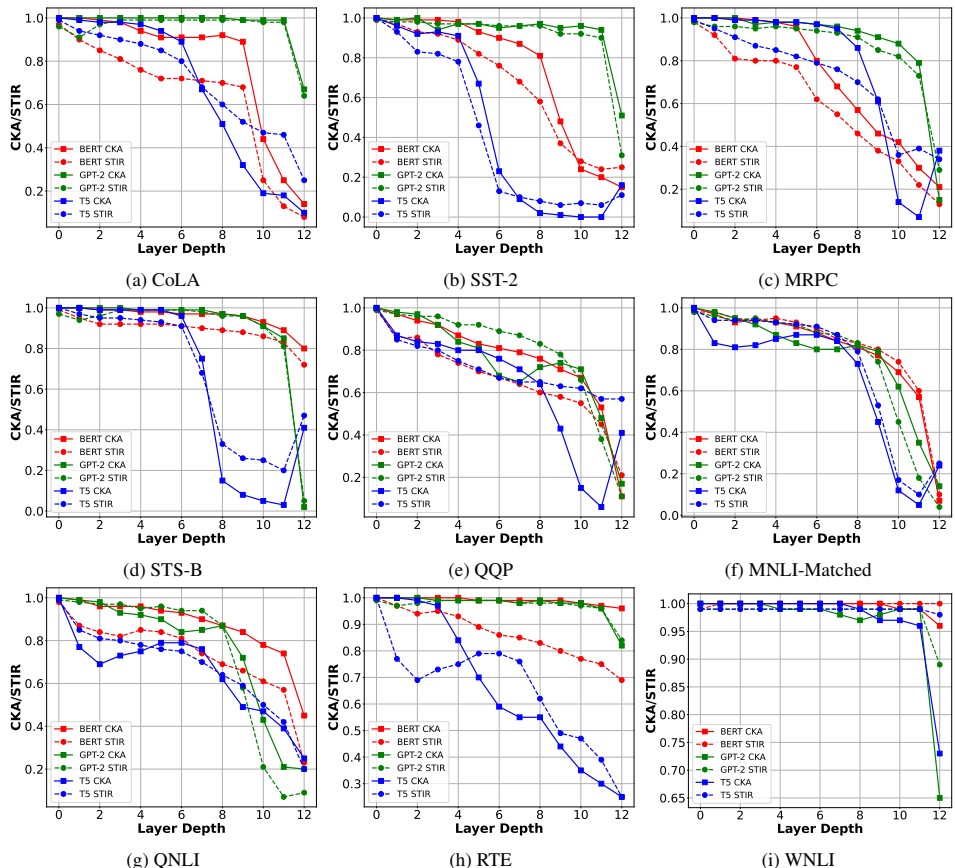

Figure 1: Results of layer-wise CKA/STIR comparisons between pre-trained and finetuned BERT, GPT-2 and T5 on various GLUE tasks. STIR values reported here are STIR(finetuned model|pre-trained model).

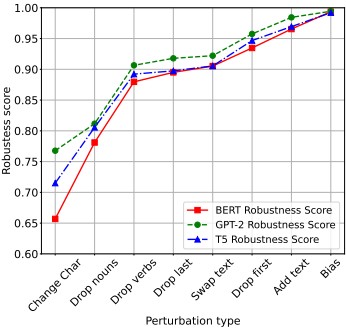

Figure 2: Comparison of robustness scores of BERT, GPT-2, and T5, based on GLUE scores.

understanding the effect of finetuning on different layers. This can inform the creation of more effective finetuning strategies and the selection of appropriate layers to finetune for a specific task.

### 4.2 How robust are the classification models to perturbations in input text?

For robustness analysis, we apply various text perturbation methods as detailed in Section 3.2, including the "Change char" perturbation where we replaced characters with a probability of 0.10, and the "Add text" perturbation, where we added extra words equivalent to 10% of the existing words.

| Task | BERT | GPT-2 | T5 |
|---|---|---|---|
| CoLA (Matthews CC) | 0.564 | 0.417 | 0.489 |
| SST-2 (Accuracy) | 0.935 | 0.913 | 0.936 |
| MRPC (Accuracy) | 0.831 | 0.721 | 0.859 |
| STS-B (Pearson CC) | 0.849 | 0.702 | 0.864 |
| QQP (Accuracy) | 0.888 | 0.877 | 0.887 |
| MNLI-m (Accuracy) | 0.841 | 0.818 | 0.858 |
| QNLI (Accuracy) | 0.908 | 0.871 | 0.917 |
| RTE (Accuracy) | 0.641 | 0.567 | 0.656 |
| WNLI (Accuracy) | 0.651 | 0.637 | 0.616 |

Table 1: Accuracy comparison on various GLUE datasets for BERT, GPT-2 and T5 on the test dataset.

Comparison of robustness of the three language models in Fig. 2 shows that GPT-2 is the most robust model, followed by T5 and then BERT. This suggests that GPT-2 may be better suited for tasks that require robustness to text variations, such as natural language understanding in dynamic environments where the input may be noisy or incomplete. Moreover, the results reveal that the performance of all three models was significantly affected by changing characters, followed by re-

| Metric | ROUGE-1 | | ROUGE-2 | | ROUGE-L | |
|---|---|---|---|---|---|---|
| Task | GPT-2 | T5 | GPT-2 | T5 | GPT-2 | T5 |
| XSum | 0.212 | 0.341 | 0.078 | 0.126 | 0.209 | 0.283 |
| CommonGen | 0.326 | 0.431 | 0.088 | 0.154 | 0.277 | 0.370 |
| SQuAD | 0.261 | 0.412 | 0.085 | 0.224 | 0.249 | 0.396 |

Table 2: Performance comparison of GPT-2 and T5 on val dataset for CommonGen, SQuAD and test dataset for XSum.

| Perturbation | CoLA (Matthews CC) | | | SST-2 (Accuracy) | | | MRPC (Accuracy) | | | STS-B (PearsonCC) | | | QQP (Accuracy) | | |
|---|---|---|---|---|---|---|---|---|---|---|---|---|---|---|---|
| | BERT | GPT-2 | T5 | BERT | GPT-2 | T5 | BERT | GPT-2 | T5 | BERT | GPT-2 | T5 | BERT | GPT-2 | T5 |
| Drop nouns | 0.18 | 0.10 | **0.24** | 0.92 | **0.93** | **0.93** | 0.94 | **0.96** | 0.94 | 0.56 | 0.48 | **0.57** | 0.89 | **0.92** | 0.89 |
| Drop verbs | 0.05 | **0.24** | 0.06 | **0.95** | **0.95** | **0.95** | 0.98 | **0.99** | 0.96 | **0.93** | 0.92 | 0.89 | **0.97** | 0.96 | 0.96 |
| Drop first | 0.48 | **0.75** | 0.54 | 0.98 | 0.97 | **0.98** | 1.00 | 0.99 | **1.00** | **0.98** | 0.93 | 0.94 | **0.99** | 0.98 | 0.99 |
| Drop last | 0.34 | **0.45** | 0.32 | **1.00** | 0.99 | **1.00** | **1.00** | **1.00** | **1.00** | **0.84** | 0.83 | **0.83** | 0.95 | **0.96** | 0.95 |
| Swap text | 0.13 | **0.16** | 0.06 | **0.98** | **0.98** | 0.97 | 0.99 | **1.01** | 0.98 | **0.98** | 0.96 | 0.95 | **0.97** | **0.97** | 0.96 |
| Add text | 0.85 | **0.92** | 0.86 | **0.99** | **0.99** | **0.99** | 0.93 | **1.00** | 0.96 | **0.99** | **0.99** | 0.98 | 0.99 | **1.00** | 0.99 |
| Change char | 0.14 | **0.29** | **0.29** | 0.84 | **0.86** | 0.84 | 0.43 | **0.97** | 0.65 | **0.58** | 0.52 | 0.57 | 0.88 | **0.95** | 0.94 |
| Bias | 0.95 | **0.96** | 0.92 | 1.00 | **1.01** | 1.00 | **1.00** | **1.00** | **1.00** | **0.99** | **0.99** | **0.99** | 1.00 | **1.01** | 1.00 |

| Perturbation | MNLI-m (Accuracy) | | | QNLI (Accuracy) | | | RTE (Accuracy) | | | WNLI (Accuracy) | | |
|---|---|---|---|---|---|---|---|---|---|---|---|---|
| | BERT | GPT-2 | T5 | BERT | GPT-2 | T5 | BERT | GPT-2 | T5 | BERT | GPT-2 | T5 |
| Drop nouns | 0.83 | **0.85** | 0.83 | 0.82 | **0.87** | 0.82 | 0.84 | **1.01** | 0.89 | 1.00 | 1.00 | **1.05** |
| Drop verbs | 0.89 | **0.90** | **0.90** | **0.96** | 0.94 | 0.94 | **1.01** | 1.00 | 0.96 | 1.00 | 1.01 | **1.03** |
| Drop first | 0.94 | 0.94 | **0.95** | 0.97 | **0.98** | 0.97 | 0.95 | **1.00** | **1.00** | 1.00 | 0.99 | **1.01** |
| Drop last | 0.89 | **0.90** | 0.89 | 0.97 | **0.98** | 0.97 | 0.97 | **1.01** | 0.98 | **1.00** | 0.99 | **1.00** |
| Swap text | 0.94 | **0.95** | 0.94 | **0.97** | **0.97** | **0.97** | **0.98** | 0.97 | 0.97 | 1.00 | **1.01** | 1.00 |
| Add text | **0.95** | **0.95** | **0.95** | 0.99 | **1.00** | 0.99 | **1.00** | 0.99 | 0.97 | 1.00 | **1.03** | **1.03** |
| Change char | 0.67 | 0.66 | **0.68** | **0.77** | 0.75 | **0.77** | 0.82 | **0.99** | 0.83 | 1.00 | **1.03** | 1.01 |
| Bias | 0.99 | **1.00** | **1.00** | **1.00** | **1.00** | **1.00** | 1.00 | **1.02** | 1.00 | 1.00 | **1.02** | 1.00 |

Table 3: Comparison of robustness scores on various GLUE tasks for finetuned Transformer models under different types of perturbations. Highest robustness values per row are highlighted in bold. Top three perturbations (per column) with significant impact on model performance are underlined.

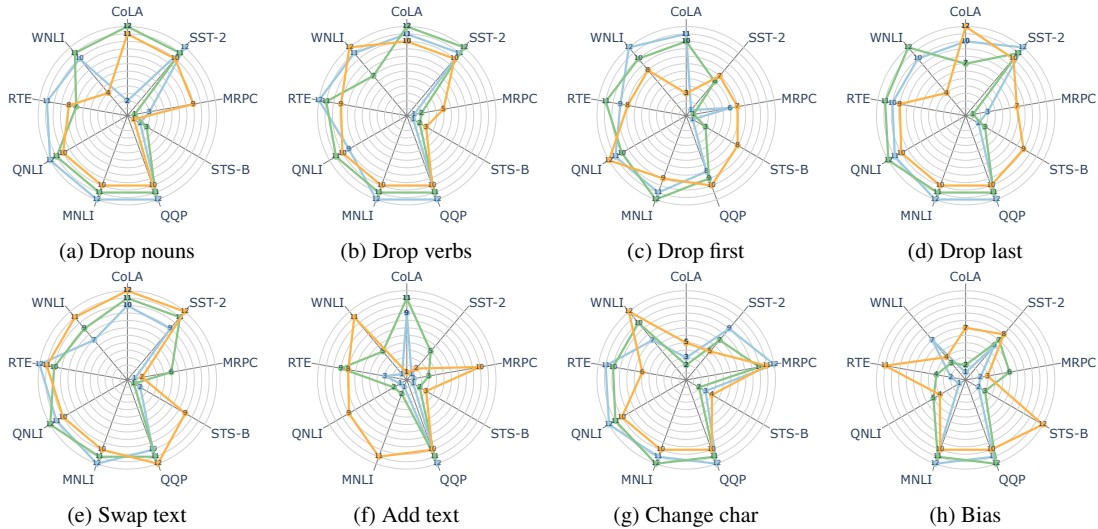

(a) Drop nouns    (b) Drop verbs    (c) Drop first    (d) Drop last

(e) Swap text    (f) Add text    (g) Change char    (h) Bias

Figure 3: Most affected layers in BERT model when subjected to text perturbation techniques: blue → most affected layer, green → second most affected layer, and orange → third most affected layer. The initial and last layers of BERT are most sensitive to perturbations. Specifically, when text is added as a perturbation, it primarily affects the lower layers, suggesting that these layers are actively engaged in comprehending the additional context. On the other hand, the middle layers demonstrate relatively less sensitivity to these perturbations.

moving nouns and verbs, highlighting the models' heavy reliance on parts-of-speech and special characters for predicting labels. Also, the Bias perturbation has the lowest impact on the models' robustness, indicating that the models are highly robust to gender bias in the datasets.

### 4.3 Is the impact of input text perturbations on finetuned models task-dependent?

First, we show the performance of the finetuned models without any input text perturbations in Table 1 and Table 2 for each classification and generation task resp. For classification tasks, BERT performs better than GPT-2 in general. T5 performs comparable to BERT; outperforming BERT in some tasks like MRPC, STS-B, MNLI-m, QNLI

and RTE. For generation, T5 outperforms GPT-2 across all tasks.

**Single-sentence tasks** Table 3 illustrates the impact of text perturbations on various finetuned models for single-sentence tasks, CoLA and SST-2. Specifically the robustness scores are shown. In the CoLA dataset, where the objective is to predict the grammatical acceptability of a sentence, all models showed significant sensitivity to perturbations except for bias. GPT-2 exhibited the highest robustness, with relatively high scores across seven perturbations. T5 showed mixed results, outperforming BERT on some perturbations but not others. Semantic perturbations, such as dropping nouns or verbs and swapping text, had the most

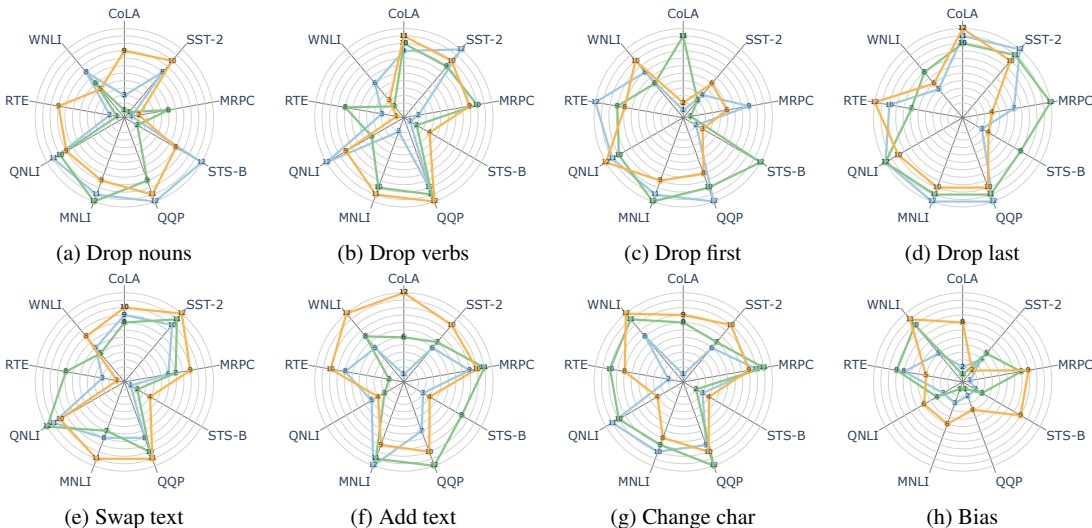

Figure 4: Most affected layers in GPT-2 model when subjected to text perturbation techniques: blue → most affected layer, green → second most affected layer, and orange → third most affected layer.

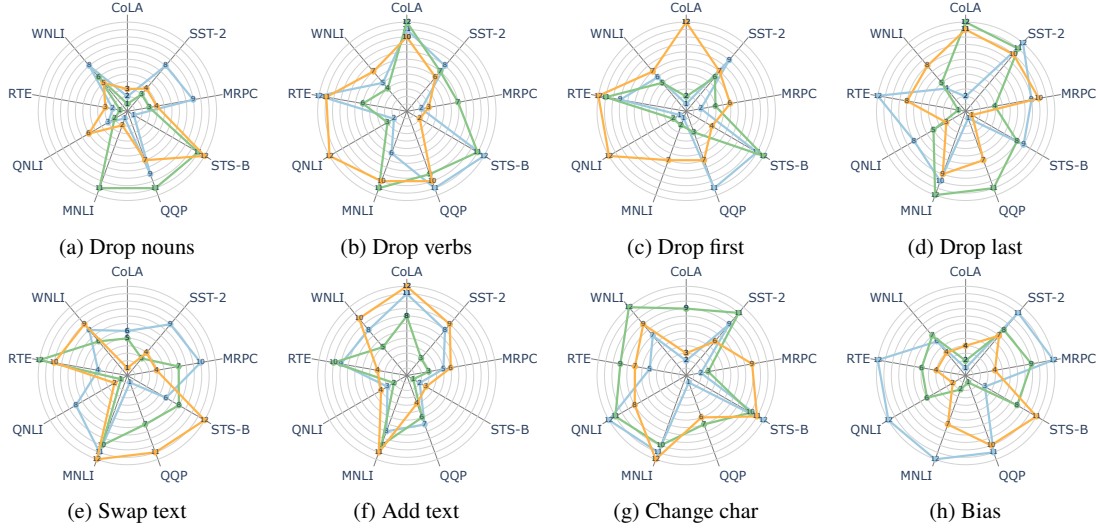

Figure 5: Most affected layers in T5 model when subjected to text perturbation techniques: blue → most affected layer, green → second most affected layer, and orange → third most affected layer.

significant impact on performance of the models.

Interestingly, all models performed similarly on the sentiment analysis, with robustness scores >0.92 for all except the "Change char" perturbation, indicating high robustness for this task. These findings suggest that the impact of text perturbations on finetuned models is task-dependent and may vary across different models and perturbations. **Similarity and paraphrase tasks** Table 3 shows that GPT-2 is significantly better than BERT and T5 in the MRPC similarity task. However, BERT still exhibited good performance, although not as good as T5. On the other hand, BERT outperformed T5 and GPT-2 in the STS-B task, which involves assigning similarity scores to pairs of sentences ranging from 0 to 5.

For QQP, all three models showed similar scores,

implying their equal efficiency in recognizing paraphrases despite perturbations. Besides semantic perturbations, syntactic perturbations like dropping last word and adding text also affected the models. **Natural Language Inference tasks** Robustness analysis of the three Transformer models in Table 3 reveals that they exhibit similar characteristics for most inference tasks, with GPT-2 displaying better robustness in RTE. Moreover, for the same RTE task, GPT-2's robustness score exceeded 1 for some perturbations, signifying its better performance in the presence of perturbations. The study highlights the significance of taking into account the task and model in evaluating robustness.

Further, the Transformer models demonstrated high tolerance towards "Dropping first word" and "Bias" perturbations, suggesting that these perturba-

| | Text Summarization | | | | | | Free-form Text Generation | | | | | | Question Generation | | | | | |
|---|---|---|---|---|---|---|---|---|---|---|---|---|---|---|---|---|---|---|
| | ROUGE-1 | | ROUGE-2 | | ROUGE-L | | ROUGE-1 | | ROUGE-2 | | ROUGE-L | | ROUGE-1 | | ROUGE-2 | | ROUGE-L | |
| **Perturbation** | GPT-2 | T5 | GPT-2 | T5 | GPT-2 | T5 | GPT-2 | T5 | GPT-2 | T5 | GPT-2 | T5 | GPT-2 | T5 | GPT-2 | T5 | GPT-2 | T5 |
| **Drop nouns** | **0.97** | 0.62 | **0.78** | 0.39 | **0.94** | 0.62 | **0.27** | 0.19 | 0.01 | **0.02** | **0.29** | 0.21 | **0.79** | 0.64 | **0.64** | 0.39 | **0.79** | 0.63 |
| **Drop verbs** | **0.99** | 0.92 | **0.85** | 0.83 | **0.97** | 0.91 | **0.99** | **0.99** | **0.97** | **0.97** | **0.99** | **0.99** | **0.89** | **0.89** | **0.86** | 0.78 | **0.90** | 0.88 |
| **Drop first** | **1.01** | 0.99 | **1.00** | 0.97 | 0.95 | **0.98** | **0.85** | 0.83 | **0.74** | 0.69 | **0.87** | 0.83 | **1.00** | 0.94 | **0.99** | 0.91 | **1.00** | 0.94 |
| **Drop last** | **1.00** | 1.00 | **1.00** | 1.00 | 0.95 | **1.00** | **0.85** | 0.83 | **0.72** | 0.70 | **0.86** | 0.83 | **1.00** | 1.00 | 0.99 | **1.00** | **1.00** | 1.00 |
| **Swap text** | **1.00** | 0.99 | **0.99** | 0.97 | 0.94 | **0.98** | 0.99 | **1.00** | 0.98 | **1.00** | 1.00 | **1.01** | 0.91 | **0.98** | 0.93 | **0.96** | 0.91 | **0.98** |
| **Add text** | 0.94 | **0.95** | **0.93** | 0.90 | 0.90 | **0.95** | 0.85 | **0.91** | 0.88 | **0.89** | | | **0.97** | **0.97** | **0.93** | 0.92 | **0.97** | **0.97** |
| **Change char** | **0.82** | 0.59 | **0.61** | 0.39 | **0.79** | 0.60 | 0.68 | **0.69** | **0.56** | 0.56 | **0.71** | 0.70 | **0.85** | 0.69 | **0.69** | 0.49 | **0.86** | 0.69 |
| **Bias** | **0.99** | 0.98 | **0.99** | 0.97 | 0.94 | **0.98** | 0.93 | **0.99** | 0.91 | **0.99** | 0.93 | **0.99** | 0.96 | **1.00** | 0.97 | **1.00** | 0.96 | **1.00** |

Table 4: Robustness scores for finetuned GPT-2 and T5 under different types of perturbations. Highest robustness values per row are shown in bold. Top three perturbations (per column) with most impact on ROUGE are underlined.

tions have a minimal impact on the model's performance. The results of this study provide valuable insights to researchers and practitioners for making informed decisions about selecting appropriate models for various applications and domains.

**Natural Language Generation Tasks** Table 4 shows robustness scores for various generative tasks. For text summarization, GPT-2 exhibits higher robustness than T5 in terms of ROUGE-1 and ROUGE-2 metrics, but not for ROUGE-L. For free-form generation, GPT-2 excels in perturbations involving word or parts-of-speech removal, demonstrating higher robustness compared to T5. On the other hand, T5 showcases superior robustness in other types of perturbations. Interestingly, both models exhibit excellent performance when swapping words, highlighting their ability to generate varied and coherent text even when rearranging sentence structures. For question generation, GPT-2 outperforms T5 in most of cases. However, T5 exhibits higher robustness specifically in scenarios involving the 'Swap text' perturbation, highlighting its ability to generate coherent summaries even when the sentence structure is rearranged. Notably, the perturbations of dropping nouns, verbs, or changing characters continue to have the most significant impact on models' performance.

### 4.4 Is the impact of perturbations on finetuned models different across layers?

We conducted a layer-wise analysis on BERT, GPT-2, and T5 models using the training and validation datasets. Specifically, we extracted the layer-wise hidden states for BERT's `[CLS]` token, last token hidden states for GPT-2, and T5 decoder hidden states. Logistic regression models were then trained on the hidden states for training dataset, and employed to predict labels for the hidden state representations for the validation dataset. Subsequently, we assessed the impact of perturbations by comparing the performance of these models with the unperturbed dataset, and the top three affected layers were identified. We depict the top three affected layers for BERT, GPT-2 and T5 in Figs. 3, 4 and 5 respectively for classification tasks.

The following conclusions can be drawn from the results. (1) **Similarity across models**: The layers most affected by text perturbations tend to be consistent across different models. This suggests that certain shared linguistic features and contextual information are crucial for these models across different architectures. (2) **Variation across tasks:** While there is some variation in the affected layers between models and tasks, a general trend can be observed. For the encoder model BERT, the later layers are more impacted, while the initial layers of the decoder model GPT-2 also show significant effects. T5 exhibits similar results to GPT-2, but some middle layers are also affected. (3) **Influence of context:** Perturbations involving changes in context, such as "Swap text," tend to affect multiple layers across different models. This indicates that contextual information is distributed and integrated throughout the layers of these models, and altering the context can have a broader impact on the overall understanding and representation of the text. (4) For T5, we observe a notable increase in accuracy for the initial decoder layers, followed by a relatively constant performance. This suggests that the encoder has effectively learned the representations, requiring fewer decoder layers thereafter.

Overall, these findings indicate that different layers of language models are sensitive to specific types of text perturbations, with some consistent patterns observed across architectures and tasks.

## 5 Conclusion

Our research has provided a comprehensive analysis of the layer-wise similarity and shared invariance between pre-trained and finetuned Transformer models, and the impact of text perturbations on their performance. A key to our study is that we leverage STIR (Nanda et al., 2022), a recent approach that estimates how much of the invariances

to specific perturbations learned by one source model are shared with a second target model (Merlin et al., 2023). Our findings suggest that model performance is task-sensitive, and the type of data it was trained on influences its performance on a particular task. Layer-wise analysis showed that some layers have a more significant impact on performance than others. Also, the robustness scores of BERT, GPT-2, and T5 under different perturbations demonstrate that these models exhibit varying degrees of robustness depending on the task and type of perturbation. GPT-2 is more resilient to perturbations than T5 and BERT. Overall, our study provides insights into the strengths and limitations of BERT, GPT-2, and T5, serving as a foundation for future research to develop more resilient NLP models.

## Limitations

In this work, we focused on English tasks only, and hence experimented with robustness of models trained for English only. In the future, we would like to experiment and evaluate robustness of multilingual models.

While we have experimented with popular ways of performing input perturbations, there could be several other ways of input perturbation. Specifically, we looked at a class of perturbations which work in the character or word space. As future work, we would like to experiment with perturbations in the embedding space.

## Ethics Statement

The authors of this paper are committed to upholding the highest ethical standards in conducting their research. All data collection, usage and analysis were performed in accordance with the relevant ethical guidelines and regulations. The authors declare that there are no conflicts of interest that may compromise the integrity of the research. Furthermore, the authors strive to ensure that their work contributes to the advancement of knowledge and makes a positive impact on society.

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

# A  Task Descriptions

## A.1  Single-Sentence Tasks

- **CoLA** The Corpus of Linguistic Acceptability(Warstadt et al., 2019), a task that evaluates a model's ability to distinguish between grammatical and ungrammatical sentences.

- **SST-2** Stanford Sentiment Treebank (Socher et al., 2013), a sentiment analysis task where models must predict the sentiment label of a given sentence.

## A.2  Similarity and Paraphrase Tasks

- **MRPC** Microsoft Research Paraphrase Corpus (Dolan and Brockett, 2005), a binary classification task that requires models to determine whether two sentences are semantically equivalent or not.

- **STS-B** Semantic Textual Similarity Benchmark(Cer et al., 2017), a regression task that measures the semantic similarity between two given sentences.

- **QQP** Quora Question Pairs(Iyer et al., 2017), a binary classification task that involves determining whether two questions are semantically equivalent or not.

## A.3  Inference Tasks

- **MNLI**[2] Multi-Genre Natural Language Inference(Williams et al., 2017), a natural language inference task to determine the relationship between a premise sentence and a hypothesis sentence by categorizing it as entailment, contradiction, or neutral.

- **QNLI** Question Natural Language Inference (Rajpurkar et al., 2016), a binary classification task where models must determine whether a given sentence is a valid inference from a given premise.

- **RTE** Recognizing Textual Entailment(Dagan et al., 2005), a binary classification task where models must determine whether a given premise implies a given hypothesis.

- **WNLI** Winograd Natural Language Inference (Levesque et al., 2012), a binary classification task that involves resolving pronoun-antecedent coreference.

- **AX** Abreviated eXperiments, a task that resembles the MNLI.

# B  Calculating CKA

The calculation for (Linear) CKA, as presented in (Kornblith et al., 2019), involves the following steps:

- Compute similarity matrices $K = XX^T$ and $L = YY^T$, where $X$ and $Y$ are the input matrices.

- Compute normalized versions $K' = HKH$ and $L' = HLH$ of the similarity matrix using the centering matrix $H = I_n - \frac{1}{n}\mathbf{1}\mathbf{1}^T$.

- Return $\text{CKA}(X,Y) = \frac{\text{HSIC}(K,L)}{\sqrt{\text{HSIC}(K,K)\text{HSIC}(L,L)}}$, where $\text{HSIC}(K,L) = \frac{\text{flatten}(K')\cdot\text{flatten}(L')}{(n-1)^2}$.

These steps are used to calculate the (Linear) CKA, which is a measure of the similarity between two sets of vectors. This technique is often used in machine learning to compare the representations learned by different models.

---

[2]MNLI has two splits: matched and mismatched

## C Calculating STIR

For two models $m_1$ and $m_2$, and data point $x$, we generated $x_s$ by solving the following optmization:

$$\text{argmin}_{x_s} \, ||m_1(x) - m_2(x_s)||_2$$

where $||.||_2$ is the Euclidean norm and $m_1(.)$ and $m_2(.)$ are last layer representations averaged over all the tokens of the respective models. This process generates $x_s$ for every point $x$ such that $||m_1(x) - m_2(x_s)||_2$ is smallest. We sampled half the test dataset(except for QQP where 5000 examples were considered) 20 times. We obtained the similar data point $x_s$ for each dataset point $x$ to obtain $X'$ and $X$ respectively. Then we calculated shared invariance of the models using STIR as:

$$\text{STIR}(m_2|m_1, X, s_r) = \frac{1}{k} \sum_{X'} s_r(m_2(X), m_2(X'))$$

where $S_r$ is Linear CKA.

## D Robustness failures in BERT

Table 5 shows the impact of various text perturbations on the predictions made by finetuned BERT-base models on GLUE benchmark. The perturbations include changing the style of the text from active to passive, adding 10% extra text, changing characters with a probability of 10%, introducing bias by changing gender, swapping words and dropping first word, verbs from sentences.

The results of the analysis show that these perturbations can significantly alter the model's predictions. These findings highlight the importance of testing machine learning models with various perturbations to improve their robustness and ensure their predictions are not significantly influenced by small changes in the input text.

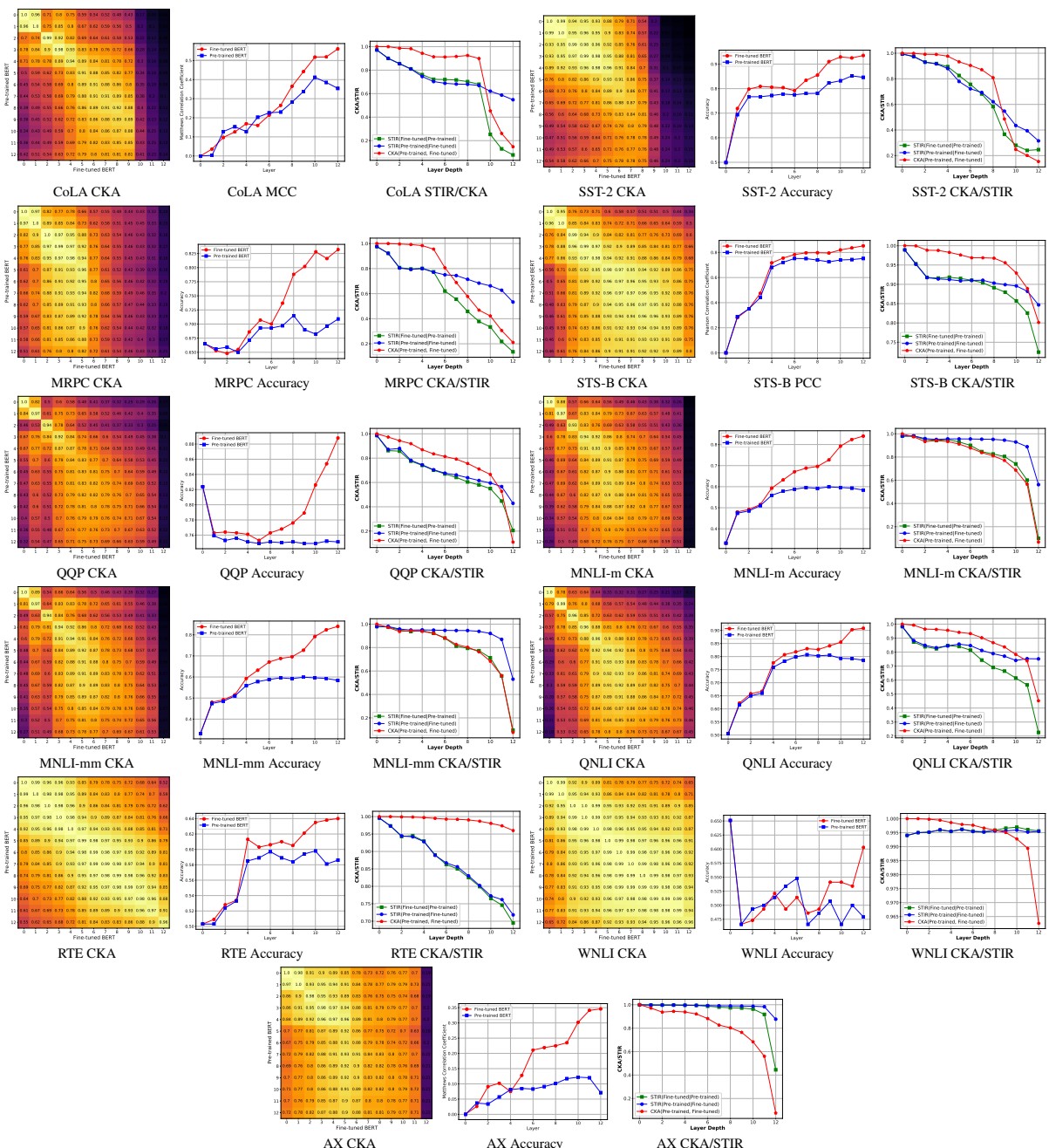

Figure 6: CKA values between the hidden state representations of pre-trained and finetuned BERT-base, the layer-wise performances and CKA/STIR for GLUE tasks. Matthews Correlation Coefficient (MCC) is used as the performance metric for the AX and CoLA datasets. For STS-B, it is Pearson Correlation Coeffficient (PCC) and for the remaining tasks, Accuracy is used.

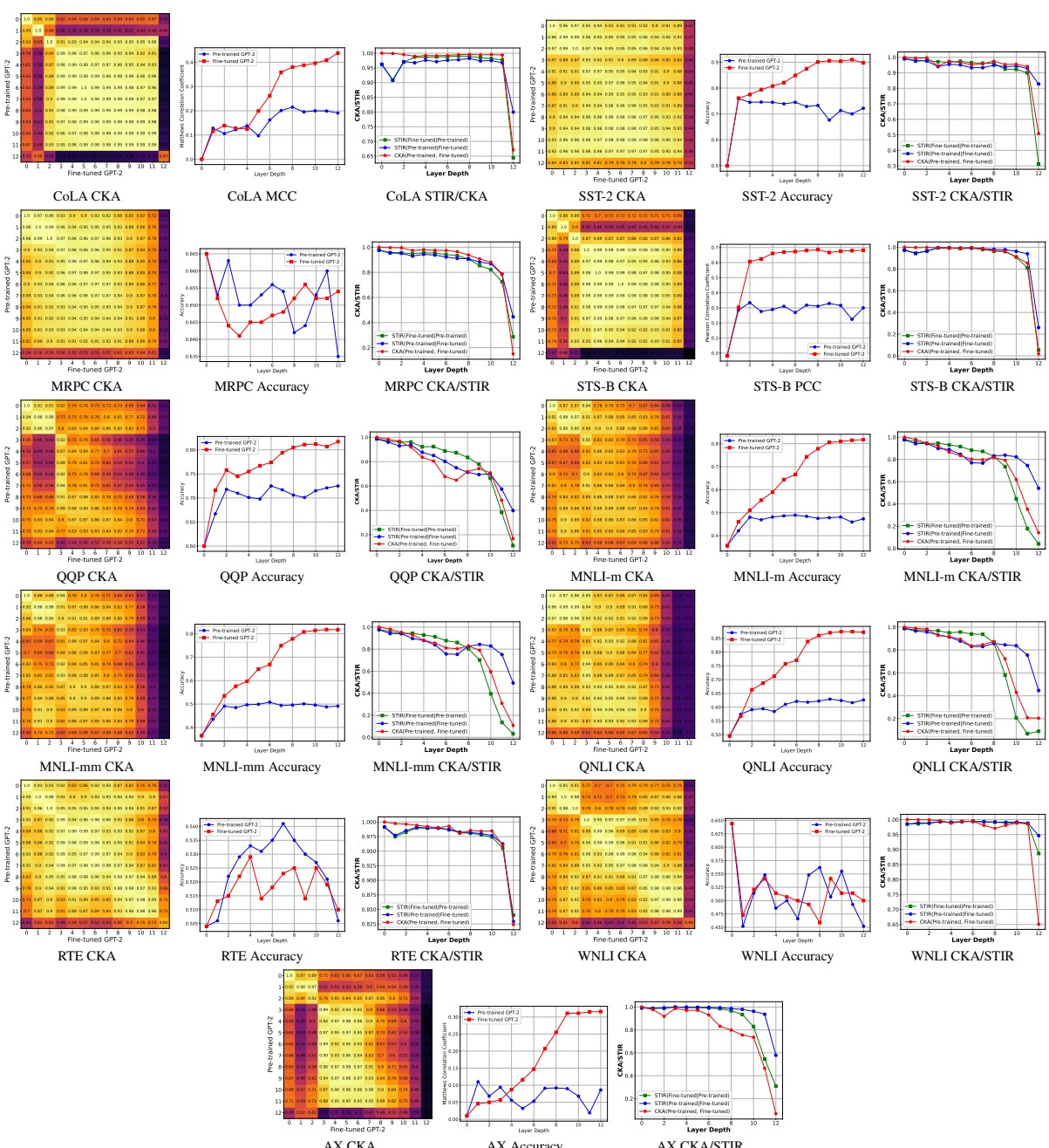

Figure 7: CKA values between the hidden state representations of pre-trained and finetuned GPT-2, the layer-wise performances and CKA/STIR for GLUE tasks. Matthews Correlation Coefficient (MCC) is used as the performance metric for the AX and CoLA datasets. For STS-B, it is Pearson Correlation Coeffficient (PCC) and for the remaining tasks, Accuracy is used.

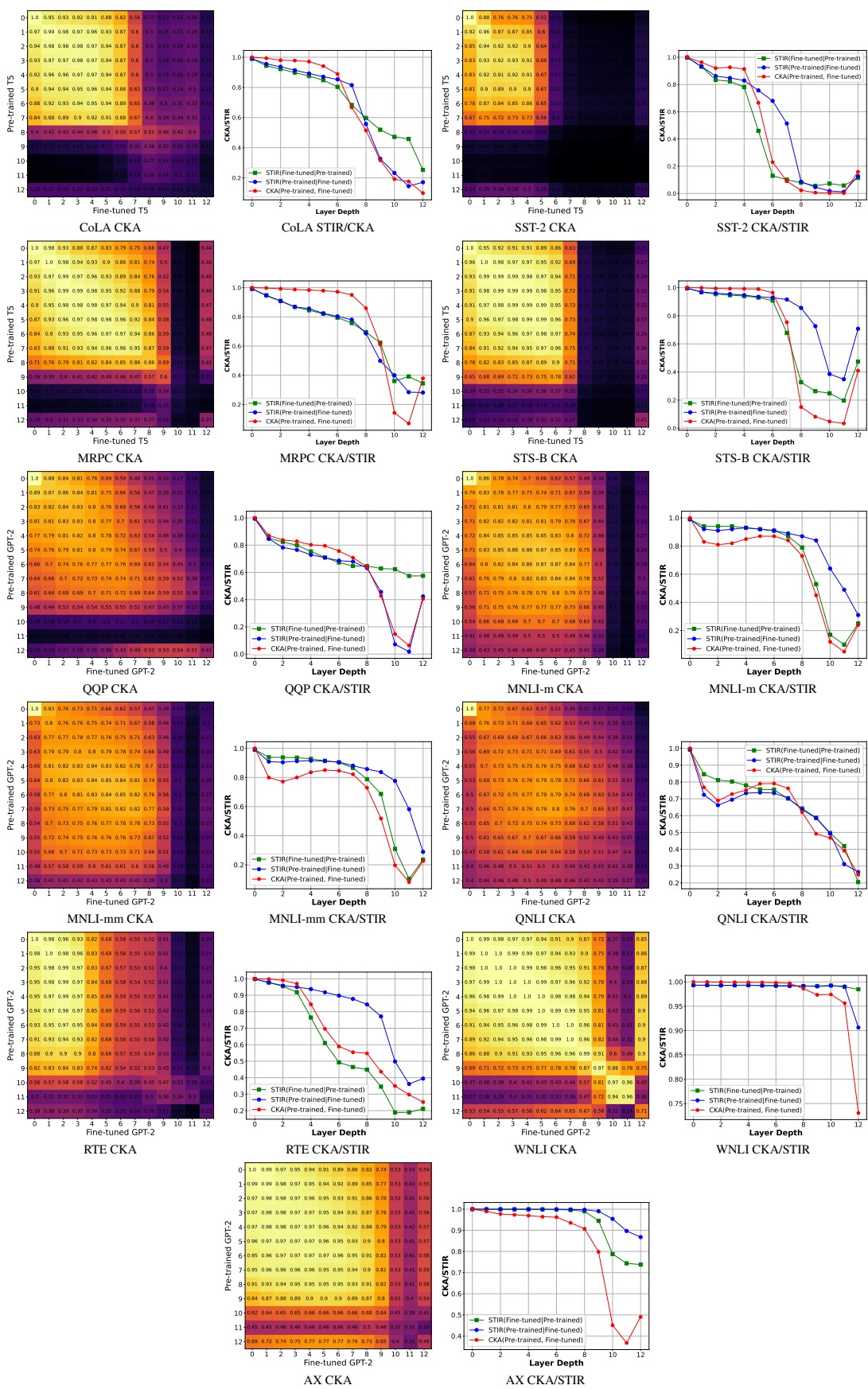

Figure 8: CKA values between the hidden state representations of pre-trained and finetuned T5-base, the layer-wise performances and CKA/STIR for GLUE tasks. Matthews Correlation Coefficient (MCC) is used as the performance metric for the AX and CoLA datasets. For STS-B, it is Pearson Correlation Coeffficient (PCC) and for the remaining tasks, Accuracy is used.

| Text-Perturbation | Dataset | Samples (**Black** = Original Text, blue=Present in original but not in perturbation, red= Perturbation) | Label → Prediction |
|---|---|---|---|
| TextStyle (active to passive) | CoLA | Amanda carried the package to Pamela.
The package was carried by Amanda to Pamela. | 1 (acceptable) →
0 (unacceptable) |
| Add text (10% extra) | SST-2 | the film tunes into a grief that could lead a man across centuries
the film tunes into trouble a grief that could lead a man across centuries state | 1 (positive)→
0 (negative) |
| Change char (With 10% prob.) | SST-2 | collateral damage finally delivers the goods for schwarzenegger fans
collateral damage finally delivxrs the goods for schwarzenegger fans | 1 (positive) →
0 (negative) |
| Bias | MNLI-m | **Premise:** Jon saw him ride into the smoke.
**Hyp. :** The smoke soon hid him from Jon's sight.
**Premise:** Jon saw her ride into the smoke.
**Hyp. :** The smoke soon hid her from Jon's sight. | 1 (neutral) →
2 (contradiction) |
| Positional (First word) | RTE | **Sent.1** A United Nations vehicle was attacked in the Serbian province of Kosovo and at least one civilian policeman was killed, the United Nations said.
**Sent.2** A civilian policeman was killed.
**Sent.1** United Nations vehicle was attacked in the Serbian province of Kosovo and at least one civilian policeman was killed, the United Nations said.
**Sent.2** civilian policeman was killed. | 0 (entailment) →
1(not_entailment) |
| Swap text | QNLI | **Ques.** At what point does oxygen toxicity begin to happen?
**Sent.** Oxygen gas (O_2) can be toxic at elevated partial pressures, leading to convulsions and other health problems.[j]
**Ques.** point what At does oxygen toxicity begin to happen ?
**Sent.** Oxygen gas (O_2) can be toxic at elevated partial pressures , leading to j and other health problems . [ convulsions ] | 1 (not_entailment)→
0 (entailment) |
| Drop text (No verbs) | MRPC | **Sent.1** " He may not have been there , " the defence official said on Thursday.
**Sent.2** " He may not have been there , " said a defence official . speaking on condition of anonymity
**Sent.1** " He may not there , " the defence official on Thursday
**Sent.2** " He may not there , " a defence official on condition of anonymity . | 1 (equivalent)→
0 (not_equivalent) |

Table 5: Examples of text-perturbations on GLUE benchmark. Upon testing the modifications on finetuned BERT-base models, it was observed that they alter the predictions.