# OpenReview forum: "On Robustness of Finetuned Transformer-based NLP Models"
_EMNLP/2023/Conference — EMNLP 2023 Findings_

### Official Review · Reviewer_C8jx · 2023-07-23

**Soundness:** 3

**Excitement:**

3: Ambivalent: It has merits (e.g., it reports state-of-the-art results, the idea is nice), but there are key weaknesses (e.g., it describes incremental work), and it can significantly benefit from another round of revision. However, I won't object to accepting it if my co-reviewers champion it.

**Paper Topic And Main Contributions:**

This paper explores the robustness of finetuned Transformer-based NLP models. The authors aim to answer three main questions: (i) Is the effect of finetuning consistent across all models for various NLP tasks? (ii) To what extent are these models effective in handling input text perturbations? and (iii) Do these models exhibit varying levels of robustness to input text perturbations when finetuned for different NLP tasks? To address these questions, the authors use two metrics to characterize changes between pretrained and finetuned language model representations across layers, and study the robustness of three popular language models (BERT, GPT-2, and T5) with various text perturbations on classification and generation tasks.
The key contributions are a systematic study of the robustness of finetuned Transformer-based NLP models to various types of input perturbation, and an analysis of the effect of finetuning on the robustness of these models.

**Questions For The Authors:**

For model robustness,  I would be more concerned with the degree of models' performance changes when using new data versus training data, not just dropping nouns, and verbs. Thus, I think comparing how the model performs on new data is also an essential part.

**Reasons To Accept:**

1. This paper presents some interesting analyses such as the robustness of transformer-based models that are affected by dropping nouns/verbs or input perturbations.
2. Layer-wise finetuning analysis can give us some insights that could select some layers to fine-tune for a specific task.


**Reasons To Reject:**

1. The paper only studies three popular language models (BERT, GPT-2, and T5), how about the effect on the large language model, such as LLaMA 7B/13B? I think the authors should conduct further experiments to show the impact of input text perturbations or dropping words.
2. The authors said that GPT was more robust than BERT/T5 across multiple types of input perturbation and the experiment supported the point, but why GPT was more robust than BERT/T5? I'd like to know the reasons and the author should further explore it.

**Reproducibility:**

3: Could reproduce the results with some difficulty. The settings of parameters are underspecified or subjectively determined; the training/evaluation data are not widely available.

**Reviewer Confidence:**

4: Quite sure. I tried to check the important points carefully. It's unlikely, though conceivable, that I missed something that should affect my ratings.

---

> ### Author Rebuttal · Authors · 2023-08-28
>
> *We value the reviewer's insights and time spent evaluating our work. Your feedback is greatly appreciated. We are grateful for your thorough reviews, as they aid us in refining and improving our research. Thank you for acknowledging the interesting aspects of our analyses, particularly the examination of transformer-based models' robustness and the insights gained from layer-wise fine-tuning analysis.*
>
> **Q: The paper only studies three popular language models (BERT, GPT-2, and T5), how about the effect on the large language model, such as LLaMA 7B/13B? I think the authors should conduct further experiments to show the impact of input text perturbations or dropping words.**
>
> * The main motivation is to consider the different categories of Transformer-based NLP model architectures like encoder (BERT), decoder (GPT-2), and encoder-decoder (T5) and cover the robustness experiments related to these three categories as all the types of the large language models belong to one of these three categories.
> * Our hypothesis is that we also expect similar insights in other large language models. As you suggested, we will perform experiments with LLaMA and Falcon models and report the results in the revised draft.
>
> **Q: The authors said that GPT-2 was more robust than BERT/T5 across multiple types of input perturbation and the experiment supported the point, but why GPT-2 was more robust than BERT/T5? I'd like to know the reasons and the author should further explore it.**
>
> Thanks for the suggestion. In this work, our goal was to make preliminary comparisons in an empirical way, and report our findings. Some possible reasons could be as follows:
> * GPT-2 has a simpler objective than BERT and T5, which means it does not need to learn additional skills such as masking.
> * GPT-2 has a larger and more diverse pre-training data, which means it can learn more generalizable and robust representations.
> * GPT-2 is a language generation model. Language generation models have been observed to exhibit more robustness to perturbations compared to classification models because they focus on generating coherent responses rather than precisely identifying class labels.
> * The next word prediction objective allows GPT-2 to capture a wide range of language patterns and makes it less likely to get overly influenced by small perturbations. MLM focuses on understanding fine-grained details may make them more vulnerable to perturbations that change the context or meaning of individual words.
>
> That said, we look forward to further systematically exploring the reasons of higher robustness of GPT-2 vs BERT/T5.
>
> **Q: For model robustness, I would be more concerned with the degree of models' performance changes when using new data versus training data, not just dropping nouns, and verbs. Thus, I think comparing how the model performs on new data is also an essential part.**
>
> * Using text perturbations as discussed in this draft does lead to "new data". By "new data", the reviewer may mean data from a different domain.
> * In that case, these models (BERT, GPT-2, T5) have been pretrained on a large variety of data across several domains including books, wikipedia, news articles, web pages, social media posts, etc.
> * Thus, it is unclear to us what the reviewer means by "new data" in this context. Kindly clarify.

---

### Official Review · Reviewer_vTiM · 2023-08-05

**Soundness:** 4

**Excitement:**

3: Ambivalent: It has merits (e.g., it reports state-of-the-art results, the idea is nice), but there are key weaknesses (e.g., it describes incremental work), and it can significantly benefit from another round of revision. However, I won't object to accepting it if my co-reviewers champion it.

**Paper Topic And Main Contributions:**

This paper conducts analyses on the impact of text perturbations of pre-trained and fine-tuned transformer models, specifically including GPT-2, T5, and BERT. In particular, this research aims at investigating whether the performance of these models vary in different tasks (such as classification, generation), and different text perturbations (such as swap text, drop noun, etc). The results show several interesting findings after conducting experiments such as task-sensitive performance, and different impact from different layers, and show the better results from GPT-2.


**Questions For The Authors:**

Why do you choose the types of text perturbations (Section 3.2)? They are the most important types, or is there any reference?

**Reasons To Accept:**

1, These analyses are useful and help to investigate more deeply into the behavior of the models on different tasks.
2, The analyses are good and broad in different aspects.


**Reasons To Reject:**

1, The findings of these results are not really significant.
2, Also, choosing different versions of a model (for instance within BERT) may also show different results and impacts.


**Reproducibility:**

3: Could reproduce the results with some difficulty. The settings of parameters are underspecified or subjectively determined; the training/evaluation data are not widely available.

**Reviewer Confidence:**

4: Quite sure. I tried to check the important points carefully. It's unlikely, though conceivable, that I missed something that should affect my ratings.

---

> ### Author Rebuttal · Authors · 2023-08-28
>
> *We value the reviewer's insights and time spent evaluating our work. Your feedback is greatly appreciated. We are grateful for your thorough reviews, as they aid us in refining and improving our research. We appreciate your recognition of the utility of our analyses in delving into model behavior across various tasks. Your observation that our analysis encompasses a wide range of aspects is encouraging.*
>
> **Q: The findings of these results are not really significant.**
>
> Our key contributions are as follows:
> -  Our analysis of finetuned models versus pretrained models shows that the last layers of the models are more affected than the initial layers when finetuning.
> - GPT-2 exhibits more robust representations than BERT and T5 across multiple types of input perturbation.
> - Although Transformers models exhibit good robustness, the models are seen to be most affected by dropping nouns, verbs or changing characters.
> - We also observed that while there is some variation in the affected layers between models and tasks due to input perturbations, certain layers are consistently impacted across different models, indicating the importance of specific linguistic features and contextual information.
>
> We believe that these are significant. We extend the line of work established by previous studies like (Wang et al., 2021; Jin et al., 2020; Li et al., 2020; Garg and Ramakrishnan, 2020; Sanyal et al., 2022).
> our study provides insights into the strengths and limitations of BERT, GPT-2, and T5, serving as a foundation for future research to develop more resilient NLP models. We will clarify these in the revised draft.
>
> **Q: Also, choosing different versions of a model (for instance within BERT) may also show different results and impacts.**
>
> * We broadly agree with this observation. That said, we expect most results on BERT to hold on variants of BERT.
> * Similarly, we expect most results on GPT-2 to hold on variants of GPT-2.
> * Architectural similarity across these variants should help preserve the general applicability of our findings. But this needs to be empirically verified.
> * We plan to do this as part of future work. We will include this in the Limitations section of the revised draft.
>
> **Q: Why do you choose the types of text perturbations (Section 3.2)? They are the most important types, or is there any reference?**
>
> * We were inspired by the recent work of Schiappa et al. (2022) [1], who have identified various types of perturbations. While there were different combinations of perturbations, our choice of eight distinct perturbations was influenced by the aim to cover the main categories of perturbations, including drop nouns, drop verbs, changing characters, etc.
>
> [1] Madeline Chantry Schiappa, Shruti Vyas, Hamid Palangi, Yogesh S Rawat, and Vibhav Vineet. 2022. Robustness analysis of video-language models against visual and language perturbations. In the 36th Conference on Neural Information Processing Systems Datasets and Benchmarks Track.

---

### Official Review · Reviewer_4B1d · 2023-08-05

**Soundness:** 3

**Excitement:**

2: Mediocre: This paper makes marginal contributions (vs non-contemporaneous work), so I would rather not see it in the conference.

**Paper Topic And Main Contributions:**

This paper proposes to analyze the robustness of pre-trained language models (encoder-based BERT, decoder-based GPT-2, encoder-decoder-based T5) on different text perturbations. The results show different models have various performance across layers on different tasks.

**Questions For The Authors:**

1. How do the authors calculate the layer-wise CKA? each layer with the first layer or the previous layer?

**Reasons To Accept:**

1. The analysis of textual perturbations is more comprehensive.

**Reasons To Reject:**

1. Although the analysis is comprehensive, the authors don't give some insight on how to fine-tune a more robust model. And how to design a more robust architecture? GPT-2 is the better one or T5? The authors can evaluate some robustness fine-tuning methods on different tasks and analysis their results.
2. This paper only focuses on text perturbations, including more robustness evaluation will be better, such as dropping some hidden states or perturbing some representations across layers.

**Reproducibility:**

3: Could reproduce the results with some difficulty. The settings of parameters are underspecified or subjectively determined; the training/evaluation data are not widely available.

**Reviewer Confidence:**

3: Pretty sure, but there's a chance I missed something. Although I have a good feel for this area in general, I did not carefully check the paper's details, e.g., the math, experimental design, or novelty.

---

> ### Author Rebuttal · Authors · 2023-08-28
>
> *We value the reviewer's insights and time spent evaluating our work. Your feedback is greatly appreciated. We are grateful for your thorough reviews, as they aid us in refining and improving our research.*
>
> **Q: Although the analysis is comprehensive, the authors don't give some insight on how to fine-tune a more robust model. And how to design a more robust architecture? GPT-2 is the better one or T5? The authors can evaluate some robustness fine-tuning methods on different tasks and analysis their results.**
>
> * In this work our goal was to study Robustness of Finetuned Transformer-based NLP Models. Specifically, in this work, we answered the following questions: (i) Is the effect of finetuning consistent across all models for various NLP tasks? (ii) To what extent are these models effective in handling input text perturbations? and (iii) Do these models exhibit varying levels of robustness to input text perturbations when finetuned for different NLP tasks?
>
> * Thank you for your suggestions. Designing a more robust finetuning architecture and exploring the comparative effectiveness of models like GPT-2 and T5 under various adversarial settings across multiple NLP tasks, will be our immediate future work.
>
> **Q: This paper only focuses on text perturbations, including more robustness evaluation will be better, such as dropping some hidden states or perturbing some representations across layers.**
>
> * Typically, perturbations are studied from the perspective of adversarial attacks. Such attacks are possible by manipulation of input information.
> * Most of the literature also performs perturbation analysis from a black box perspective where only inputs are modified.
> * Hence, we feel that dropping some hidden states or perturbing some representations across layers does not apply to our study.
>
> **Q: How do the authors calculate the layer-wise CKA? each layer with the first layer or the previous layer?**
>
> * We extract the hidden states corresponding to each token within a sentence for every language model. These hidden states are then averaged to create a unified 768-dimensional representation for that sentence. Let's consider a scenario with 100 sentences. Upon giving these sentences as input to the encoder of BERT/T5 or the decoder of GPT-2, we obtain values organized in a [13, 100, 768] dimension (comprising 1 embedding and 12 layers).
> * This methodology is applied to pre-trained and fine-tuned models, resulting in [13, 100, 768] dimensional representations for each pre-trained and fine-tuned model. Subsequently, we compute the CKA values at every layer. Specifically, we compute the CKA between the [100, 768] representations of the pre-trained and fine-tuned models. This process yields 13 layer-wise values for each combination of dataset and model.
>
> *We will clarify these in the revised draft.*

---

### Meta-Review · Area_Chair_kqLL · 2023-09-14

**Recommendation:** 3

**Metareview:**

The paper provides an empirical analysis of the robustness of fine-tuned LMs and MLMs to perturbations applied to input text.

Reviewers agree that while there is a good breadth of experiments, there’s less in terms of ablations and understanding what causes the differences in robustness (beyond speculation in the responses), which affects the level of excitement.

---

### Decision · Program_Chairs · 2023-10-07

**Decision:**

Accept-Findings

**Comment:**

The paper provides an empirical analysis of the robustness of fine-tuned LMs and MLMs to perturbations applied to input text.

Reviewers agree that while there is a good breadth of experiments, there’s less in terms of ablations and understanding what causes the differences in robustness (beyond speculation in the responses), which affects the level of excitement.